# Reshaping Diverse Planning: Let There Be Light!

**Michael Katz** and **Shirin Sohrabi**
IBM T.J. Watson Research Center
1101 Kitchawan Rd, Yorktown Heights, NY 10598, USA

### Abstract

The need for multiple plans to a planning problem has been established by various applications. In some, solution quality has the predominant role, while in others diversity is the key factor. Most recent work takes both plan quality and solution diversity into account under the generic umbrella of *diverse planning*. There is no common agreement, however, on a collection of computational problems that fall under that generic umbrella. This in particular might lead to a comparison between planners that have different solution guarantees or optimization criteria in mind. In this work we revisit diverse planning literature in search of such a collection of computational problems, classifying the existing planners to these problems. We formally define a taxonomy of computational problems with respect to both plan quality and solution diversity, extending the existing work. We propose a novel approach to diverse planning, exploiting existing classical planners via planning task reformulation and choosing a subset of plans of required size in post-processing. Based on that, we present planners for two computational problems, that most existing planners solve. Our experiments show that the proposed approach significantly improves over the best performing existing planners in terms of coverage, the overall solution quality, and the overall diversity according to various diversity metrics.

## 1  Introduction

Many applications of planning require generating multiple plans rather than one. Some examples include malware detection (Boddy et al. 2005), automated analysis of streaming data (Riabov et al. 2015), and risk management (Sohrabi et al. 2018). Planners that produce multiple plans were also found useful in the context of re-planning and plan monitoring (Fox et al. 2006), user preferences (Myers and Lee 1999; Nguyen et al. 2012), as well as the engine for plan recognition and its related applications (Sohrabi, Riabov, and Udrea 2016). All these applications justify the need for finding a *diverse* set of plans while keeping quality in mind.

Many diverse planners were developed over the last decade, each one focused on addressing a particular diversity metric. For example, while DLAMA focuses on finding a set of plans by considering a landmark-based diversity measure (Bryce 2014), LPG-d and DIV focus on finding a set of plans with a particular minimum action distance (Nguyen et al. 2012; Coman and Muñoz-Avila 2011).

Goldman and Kuter (2015) propose a diversity metric based on information retrieval literature. Roberts, Howe, and Ray (2014) suggest another diversity metric, introducing several planners, such as $itA^*$ and MQA, which, in addition to the diversity metrics, consider plan quality. Recently, Vadlamudi and Kambhampati (2016) suggested "cost-sensitive" diverse planners, first finding all cost sensitive plans and then finding a diverse set of plans among these. Top-k planners (e.g., Katz et al. 2018b) or top-quality planners (Katz, Sohrabi, and Udrea 2019b) can also be viewed as diverse planners, which find a set of plans purely addressing the quality metric.

Despite the large number of existing tools and diversity metrics, there is no adopted collection of computational problems in diverse planning, making the comparison of different approaches challenging. Further, mixing quality and diversity creates an additional challenge for comparing various planners, especially if they have different optimality guarantees. However, even for the same computational problem, planner comparison can be challenging. Every planner can have a different implementation of the same diversity metric, and many planners produce a collection of plans without specifying the metric used, or the solution value under that metric. To the best of our knowledge, there exists no external validation tool for a collection of plans, producing the solution value under a given diversity metric. Additionally, most of the diverse planning approaches compute the set of plans by repeatedly solving the same task. To obtain a different behavior, planner's heuristic guidance is modified to account for already found plans, with a specific focus on a particular metric. This requires (a) having an intimate familiarity with the way a particular planner works, and (b) creating a separate modification for each metric. However, the outcome is not always as intended. Tweaking the heuristic function does not necessarily result in a different plan and planners have to discard many equal plans and repeat unnecessary iterations.

In this work, we address the computational problems in diverse planning as well as the diverse planner construction paradigm. Similarly to the separation in classical planning, we distinguish between optimal, bounded, and satisficing diverse planning and map the existing planners to their respective categories. We propose a new quality metric for a set of plans, measuring how close the plans are to the best

subset of all known plans. We create an external validation tool for the metrics considered in this paper, allowing us to compute the diversity values of the solutions produced by existing planners. We introduce an alternative planner construction paradigm, a diverse planning algorithm that instead of modifying a planner, modifies a planning task. Following the ideas of Katz et al. (2018b), we suggest reformulating the planning task after each iteration, forbidding sets of plans. Next, we post-process the found plans to derive a subset of plans of the required size, according to the given metric. Our approach, Forbid Iterative (FI), is not restricted to any planner and can exploit the recent advances in classical planning. To demonstrate this advantage, we experiment with one of the recent best-performing approaches to agile planning, heuristic novelty of the red-black planning heuristic (Katz et al. 2017; Katz, Hoffmann, and Domshlak 2013; Domshlak, Hoffmann, and Katz 2015), a core component for several participants of the recent International Planning Competition (IPC) 2018 (Katz et al. 2018a; Katz 2018). Based on this approach, we create planners for two of the introduced computational problems. We show that the same approach outperforms the dedicated planners built for specific metrics on these metrics and on their linear combinations, for both computational problems.

## 2 Preliminaries and Related Work

A SAS$^+$ *planning task* (Bäckström and Nebel 1995) is given by a tuple $\langle \mathcal{V}, \mathcal{A}, s_0, s_* \rangle$, where $\mathcal{V}$ is a set of *state variables*, $\mathcal{A}$ is a finite set of *actions*. Each state variable $v \in \mathcal{V}$ has a finite domain $dom(v)$. A pair $\langle v, \vartheta \rangle$ with $v \in \mathcal{V}$ and $\vartheta \in dom(v)$ is called a *fact*. A (partial) assignment to $\mathcal{V}$ is called a *(partial) state*. Often it is convenient to view partial state $p$ as a set of facts with $\langle v, \vartheta \rangle \in p$ if and only if $p[v] = \vartheta$. Partial state $p$ is *consistent* with state $s$ if $p \subseteq s$. We denote the set of states of a planning task by $\mathcal{S}$. $s_0$ is the *initial state*, and the partial state $s_*$ is the *goal*. Each *action* $a$ is a pair $\langle pre(a), eff(a) \rangle$ of partial states called *preconditions* and *effects*. An *action cost* is a mapping $C : \mathcal{A} \to \mathbb{R}^{0+}$. An action $a$ is applicable in a state $s \in \mathcal{S}$ if and only if $pre(a)$ is consistent with $s$. Applying $a$ changes the value of $v$ to $eff(a)[v]$, if defined. The resulting state is denoted by $s[\![a]\!]$. An action sequence $\pi = \langle a_1, \ldots, a_k \rangle$ is applicable in $s$ if there exist states $s_0, \cdots, s_k$ such that (i) $s_0 = s$, and (ii) for each $1 \leq i \leq k$, $a_i$ is applicable in $s_{i-1}$ and $s_i = s_{i-1}[\![a_i]\!]$. We denote the state $s_k$ by $s[\![\pi]\!]$. $\pi$ is a plan iff $\pi$ is applicable in $s_0$ and $s_*$ is consistent with $s_0[\![\pi]\!]$. We denote by $\mathcal{P}(\Pi)$ (or just $\mathcal{P}$ when the task is clear from the context) the set of all plans of $\Pi$. The cost of a plan $\pi$, denoted by $C(\pi)$ is the summed cost of the actions in the plan.

The distance between two plans $\pi, \pi'$ is defined as $\delta(\pi, \pi') = 1 - \text{sim}(\pi, \pi')$, where the similarity measure sim is between 0 (two plans are unrelated) and 1 (equivalent). The diversity of a set of plans, $D(P)$, $P \subseteq \mathcal{P}$ is then defined as some aggregation (e.g., min or average) of the pairwise distance within the set $P$. While some domain-dependent similarity measures exist (e.g., Myers and Lee 1999; Coman and Muñoz-Avila 2011), recent research has focused on domain-independent measures, comparing plans based on their actions, states, causal links, or landmarks (Nguyen

et al. 2012; Bryce 2014).

Stability similarity (inverse of the plan distance (Fox et al. 2006; Coman and Muñoz-Avila 2011)) measures the ratio of the number of actions that appear on both plans to the total number of actions on these plans, referring to plans as action sets, ignoring repetitions. Given two plans $\pi, \pi'$, it is defined as $\text{sim}_{\text{stability}}(\pi, \pi') = |A(\pi) \cap A(\pi')|/|A(\pi) \cup A(\pi')|$, where $A(\pi)$ is the set of actions in $\pi$. Uniqueness similarity (Roberts, Howe, and Ray 2014) is another measure that considers plans as action sets. It measures whether two plans are permutations of each other, or one plan is a partial plan (subset) of the other plan. State similarity measures similarity between two plans based on representing the plans as a sequence of states, where each state is a set of predicates. While there are multiple ways to define state similarity, we adapt the following definition from (Nguyen et al. 2012), modifying it based on use of similarity rather than distance between plans. Let $(s_0, s_1, \ldots, s_k)$ and $(s_0, s'_1, \ldots, s'_{k'})$ be the sequences of states traversed by the plans $\pi$ and $\pi'$, respectively. Let $\Delta(s, s') = |s \cap s'|/|s \cup s'|$ be the similarity between two states. Assuming $k' \leq k$, the state similarity measure is defined as follows: $\text{sim}_{\text{state}}(\pi, \pi') = \sum_{i=1}^{k'} \Delta(s_i, s'_i) \ k$. Note, each state $s'_{k'+1}, \ldots, s_k$ is considered to not contribute to the similarity measure (i.e., zero is considered). The combination of the state and uniqueness measures address some of the major weaknesses of the stability measure raised by recent research (Goldman and Kuter 2015). Thus, since our focus in this work is not on metrics, we omit the description of the landmark-based distance (Bryce 2014).

While there seems to be no widely adopted definitions of diverse planning problems, previous work has introduced some such definitions. In these definitions, $d$ is a threshold on the distance and $c$ is a threshold of the cost of the plans. The variant introduced by Nguyen et al. (2012) requires the distance between every pair of plans in the solution to be of bounded diversity. Formally, the search problem is depicted as follows:

$$\text{dDISTANTkSET} : \text{find } P \text{ with } P \subseteq \mathcal{P}, \tag{1}$$
$$|P| = k, \min_{\pi, \pi' \in P} \delta(\pi, \pi') \geq d.$$

Another variant, by Vadlamudi and Kambhampati (2016) extends the previous search problem by requiring each individual plan in the solution to be of bounded quality. Formally:

$$\text{cCOSTdDISTANTkSET} : \text{find } P \text{ with } P \subseteq \mathcal{P},$$
$$|P| = k, \min_{\pi, \pi' \in P} \delta(\pi, \pi') \geq d, C(\pi) \leq c \ \forall \pi \in P. \tag{2}$$

While Eq. 2 considers plan costs, Eq. 1 only considers the distance between plan pairs. Note that both definitions require finding $k$ distinct plans.

We denote the diversity of a set of plans $P$, computed as an average over the pairwise dissimilarity of the set $P$, under the similarity measures of stability, uniqueness, and state by $D_a$, $D_u$, and $D_s$, respectively, dropping $P$ for readability. Also, $D_{ma}$ denotes the diversity metric computed as minimum over the pairwise stability dissimilarity.

## 3 Quality Metric

While most work in diverse planning focused on the diversity metrics, not much was done for quality metrics. One possible quality metric is the summed cost of plans $Q = \sum_{\pi \in P} C(\pi)$. In order to normalize its value, it is possible, as with the International Planning Competition (IPC) quality metric for individual plans, to divide the best known solution value by the value of the given planner. One downside of such a metric is that a single plan's quality can have a large effect on the overall quality. For example, the quality of a set of plans, where all plans are optimal except for one, of a much higher cost, may get a quality score worse than a set where all plans are not optimal. Thus, we suggest a quality metric that will allocate a score to each plan in the set, aggregating these scores into a single score.

Given $n$ diverse planners, let $P = P_1 \cup P_2 ... \cup P_n$ be the set of all plans found by these planners. Let $\pi_1, \ldots \pi_k$ be $k$ plans with the lowest cost, ordered by their cost from smallest to largest and let $c_i = C(\pi_i)$. For a planner $j$, the quality of the solution $P_j$ is measured relatively to the best known $k$ plan costs $c_1, \ldots, c_k$ as follows. Let $\pi_1^j, \ldots \pi_k^j$ be an ordering of plans in $P_j$ according to their costs and let $c_i^j = C(\pi_i^j)$. The quality metric is defined as follows.

$$Q(P_j) := \frac{1}{k} \times \sum_{i=1}^{k} \frac{c_i}{c_i^j}. \tag{3}$$

Note that $c_i^j \geq c_i$, since $P_j \subseteq P$, and thus $\pi_i^j$ has at least $i - 1$ plans of no larger cost in $P$. Thus, each sum component is between 0 and 1, and thus the whole score is a value between 0 and 1. Further, a solution $P_j$ will get the score 1 if and only if it consists of $k$ cheapest plans found by any planner. In other words, if there exists no plan in $P \setminus P_j$ (found by any of the other planners) that is cheaper than a plan in $P_j$. The suggested metric is similar in spirit to the parsimony ratio (Roberts, Howe, and Ray 2014). The parsimony ratio is defined as $s(\pi_k, \pi_l) = |\pi_k|/|\pi_l|$, where for each $\pi_l$ ($|\pi_l| = l$) we need to find an optimal plan, $\pi_k$ ($|\pi_k| = k$), such that $\pi_k \subseteq \pi_l$, $k \leq l$. This can be challenging by itself, since it requires finding optimal plans. The parsimony ratio also only considers unit cost plans. Both these limitations do not exist in our suggested metric: it can handle general costs and the computation is relative to the set of known plans.

## 4 Diverse Planning Revisited

In this section, we define a collection of computational problems in diverse planning for two optimization criteria, quality and diversity. Following previous definitions, depicted in Eqs. 1 and 2, we define a solution to a diverse planning problem as a set of plans of a required size. In contrast to previous definitions, in case there exist fewer plans than requested, the set of all plans is also considered to be a valid solution.

**Definition 1 (Diverse planning solution)** *Let* $\Pi$ *be a planning task and* $\mathcal{P}$ *be the set of all plans for* $\Pi$. *Given a natural number* $k$, $P \subseteq \mathcal{P}$ *is a* $k$*-diverse planning solution if* $|P| = k$ *or* $P = \mathcal{P}$ *if* $|\mathcal{P}| < k$.

Restricting our attention to two optimization criteria, quality and diversity, let us introduce some terminology. We say that a solution is *quality-optimal* (*diversity-optimal*) if there exists no solution of better quality (diversity). In other words, given solution quality mapping $Q$ (diversity mapping $D$), a solution $P$ is quality-optimal (diversity-optimal) if for all solutions $P'$ we have $Q(P') \leq Q(P)$ ($D(P') \leq D(P)$). Given a bound $b$, we say that a solution $P$ is *quality-bounded* (*diversity-bounded*) if $Q(P) \geq b$ ($D(P) \geq b$).

For both quality and diversity, one could either strive to find optimal or bounded solutions, or impose no restriction on solution quality. Unfortunately, these two optimization criteria can interfere with each other. Thus, in what follows, we define various search and optimization problems.

### 4.1 Satisficing Diverse Planning

We start with imposing no restrictions. Thus, the *Satisficing Diverse Planning* problem can be defined as follows.

sat-k : Given $k$, find a $k$-diverse planning solution.

Note that the objective is to find any set of $k$ plans without any restrictions on either quality or diversity. This is the category under which most diverse planners fall (e.g., Bryce 2014; Roberts, Howe, and Ray 2014). To compare planners in this category, it is sufficient to compare the quality and diversity of their solutions. Note, many of the satisficing diverse planners incorporate the distance measure into their search and focus on finding diverse plans with respect to that particular distance measure in mind. Hence, while they may perform well for one diversity metric, they may do poorly in another one.

### 4.2 Bounded Diverse Planning

Continuing now by restricting either quality or diversity by imposing a bound, we introduce a *Bounded Quality (Diversity) Diverse Planning*. We do that by restricting the set of feasible solutions.

**Definition 2 (Diversity-bounded solution)** *Let* $\Pi$ *be a planning task,* $D$ *be some diversity metric,* $b$ *be some bound, and* $\mathcal{P}$ *be the set of all* $\Pi$*'s plans. Given a natural number* $k$, $P \subseteq \mathcal{P}$ *is a* $b$*-diversity-bounded* $k$*-diverse planning solution if it is a* $k$*-diverse planning solution and* $D(P) \geq b$.

**Definition 3 (Quality-bounded solution)** *Let* $\Pi$ *be a planning task,* $Q$ *be some quality metric,* $c$ *be some bound, and* $\mathcal{P}$ *be the set of all* $\Pi$*'s plans. Given a natural number* $k$, $P \subseteq \mathcal{P}$ *is a* $c$*-quality-bounded* $k$*-diverse planning solution if it is a* $k$*-diverse planning solution and* $Q(P) \geq c$.

Given the definitions above, we can now define the following search problems:

bD-k : Given $k$ and $b$, find a $b$-diversity-bounded
$k$-diverse planning solution,

bQ-k : Given $k$ and $c$, find a $c$-quality-bounded
$k$-diverse planning solution.

Note that solutions for bQ-k can be obtained from solutions to the Top-quality planning problem (Katz, Sohrabi, and Udrea 2019b).

The search problem bD-k generalizes the definition in Eq. 1 by Nguyen et al. (2012), for a diversity score $D_{ma}$ defined as the minimum over the pairwise stability dissimilarity. Note that this measure differs from $D_a$, that averages over the pairwise stability dissimilarity. For bounded diverse planning, $D_{ma}$ dominates $D_a$ in the sense that solutions to the diversity-bounded diverse planning under $D_{ma}$ are necessarily solutions to the diversity-bounded diverse planning under $D_a$ with the same bound, but not the other way around. The planner LPG-d implements the approach of Nguyen et al. (2012), for a variant of $D_{ma}$, where the stability similarity is computed over multisets, instead of sets. We denote this diversity metric by $D_{mma}$. Thus, LPG-d can be thought of as a diversity-bounded diverse planner for $D_{mma}$ but not for any of the other metrics. Further, while LPG-d is a sound planner, it is not complete, since it can only add plans to the collection of previously found plans, and never reconsiders the decision to add a plan. Thus, in principle LPG-d might not be able to find a solution to the diversity-bounded diverse planning problem when a solution exists.

Restricting both quality and diversity results in an additional search problem, one we call *Bounded Quality and Diversity Diverse Planning*.

> bQbD-k : Given $k, b,$ and $c$, find a $c$-quality-bounded
> and $b$-diversity-bounded $k$-diverse planning solution.

The search problem bQbD-k generalizes the definition in Eq. 2 by Vadlamudi and Kambhampati (2016), for diversity score that uses min as the aggregation method and quality score defined as a maximum over the individual plan costs. It is worth noting here that in all these definitions, as in classical planning, if the bound is super-optimal, the search problem is considered to be unsolvable.

### 4.3 Optimal Diverse Planning

Restricting now either the quality or diversity to be optimal, we define two optimization problems, *Optimal Quality (Diversity) Diverse Planning*.

> optQ-k : Given $k$, find a quality-optimal
> $k$-diverse planning solution.

> optD-k : Given $k$, find a diversity-optimal
> $k$-diverse planning solution.

Top-k planners (e.g., Riabov, Sohrabi, and Udrea 2014; Katz et al. 2018b) can be viewed as planners for optQ-k, optimizing the quality metric $Q = \sum_{\pi \in P} C(\pi)$. To the best of our knowledge, there are no existing planners for the optD-k optimization problem. In fact, it is not clear how to create such non-trivial planners, without the need to generate the set of all plans.

If we further restrict the other optimization function, this results in additional optimization problems. The first two

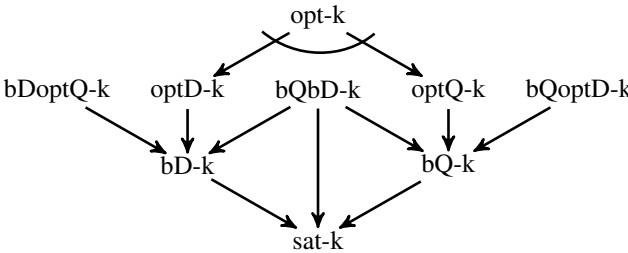

Figure 1: Hierarchy between the computational problems.

are *Optimal Quality Bounded Diversity Diverse Planning* and *Optimal Diversity Bounded Quality Diverse Planning*, as follows.

> bDoptQ-k : Given $k$ and $b$, find a quality-optimal
> among $b$-diversity-bounded $k$-diverse planning solutions.

> bQoptD-k : Given $k$ and $c$, find a diversity-optimal
> among $c$-quality-bounded $k$-diverse planning solutions.

Note that the solutions to the optimization problems bDoptQ-k and bQoptD-k are relative to the restricted set of solutions as in Definitions 2 and 3, respectively. This means that a solution to, e.g., bQoptD-k is not necessarily a solution to optD-k. One possible way to obtain solutions to the bQoptD-k optimization problem is by using a top-k planner to generate a set of all plans of bounded quality and then select an optimal subset of size $k$ from the generated set according to some diversity metric.

We can further restrict a set of feasible solutions to quality-optimal (diversity-optimal) diverse planning solutions and choose the best according to the diversity (quality) metric among those. Instead, our last optimization problem we simply call *Optimal Diverse Planning*. The objective of optimal diverse planning is to find a solution that is *pareto-optimal*, that is for all solutions $P'$ we have either $Q(P') \leq Q(P)$ and for all $P''$ with $Q(P) = Q(P'')$ we have $D(P'') \leq D(P)$ or $D(P') \leq D(P)$ and for all $P''$ with $D(P) = D(P'')$ we have $Q(P'') \leq Q(P)$. In words, optimal solutions are solutions on the pareto frontier of quality and diversity. We denote the optimization problem stated above by opt-k.

The hierarchy between the presented computational problems is depicted in Figure 1. Edges represent solution set inclusion, i.e., whether a solution for one problem is necessarily a solution for another, assuming a solution exists. For example, a pareto-optimal solution is a solution to either optD-k or optQ-k, but not necessarily to either bQoptD-k or bDoptQ-k, since the latter two optimize over the solutions that are of bounded quality and diversity, respectively. The diagram does not reflect the transitive inclusion, which, in this case, means that solutions to all problems are solutions to the satisficing diversity planning problem.

# 5 Satisficing Diverse Planning with a Satisficing Classical Planner

Previous work has focused on modifying existing planners, either heuristic search or local search based ones, to come up with plans that differ from previously found ones. These modified planners were then applied to the same planning task, over and over again. We suggest a different approach, using possibly the same planner, iteratively modifying the planning tasks to forbid plan sets (Katz et al. 2018b). Below, we list some of the benefits to such an approach: (1) it allows us to exploit state-of-the-art classical planners without the need to modify them, taking advantage of the progress in classical satisficing planning; (2) it removes the need for modifying the behaviour of the existing planners, allowing these planners to work as intended; (3) this allows us to take the selection of a subset of plans that is diverse according to a specific metric and postprocess them, thus also allowing us to define and use more sophisticated metrics.

## 5.1 Forbidding a Plan as a Multiset of Actions

Existing literature suggests one such task reformulation, forbidding exactly the given set of plans (Katz et al. 2018b). This was done in the context of top-k planning, where plans could not be safely discarded from consideration. In satisficing diverse planning, there is no such limitation. As a result, it is possible to forbid additional plans. One could envision a metric-dependent reformulation, forbidding also the plans that are similar according to the given metrics. With the stability metric in mind, we suggest a reformulation that ignores orders between actions in a plan and thus, also forbids all possible reorderings of a given plan. Below, we present the detailed description of such a reformulation.

**Definition 4** *Let $\langle \mathcal{V}, A, s_0, s_* \rangle$ be a planning task and $X$ be a multiset of actions. The task $\Pi_X^- = \langle \mathcal{V}', A', s_0', s_*' \rangle$ is defined as follows.*

- $\mathcal{V}' = \mathcal{V} \cup \{\overline{v}\} \cup \{\overline{v}_o \mid o \in X\}$, with $\overline{v}$ being a binary variable, and $dom(\overline{v}_o) = \{0, \ldots, m_o\}$, where $m_o$ is the number of occurences of $o$ in $X$,

- $A' = \{o^e \mid o \in A \setminus X\} \cup \{o^r, o^d \mid o \in X\} \cup \bigcup_{i=1}^{m_o} \{o_i^f \mid o \in X\}$, where

  $o^e = \langle pre(o), \mathit{eff}(o) \cup \{\langle \overline{v}, 0 \rangle\} \rangle$,

  $o^r = \langle pre(o) \cup \{\langle \overline{v}, 0 \rangle\}, \mathit{eff}(o) \rangle$,

  $o^d = \langle pre(o) \cup \{\langle \overline{v}, 1 \rangle, \langle \overline{v}_o, m_o \rangle\}, \mathit{eff}(o) \cup \{\langle \overline{v}, 0 \rangle\} \rangle$,

  $o_i^f = \langle pre(o) \cup \{\langle \overline{v}, 1 \rangle, \langle \overline{v}_o, i\text{-}1 \rangle\}, \mathit{eff}(o) \cup \{\langle \overline{v}_o, i \rangle\} \rangle$,

  $C'(o^e) = C'(o^r) = C'(o^d) = C'(o^f) = C(o)$,

- $s_0'[v] = s_0[v]$ *for all* $v \in \mathcal{V}$, $s_0'[\overline{v}] = 1$, *and* $s_0'[\overline{v}_o] = 0$ *for all* $o \in X$, *and*

- $s_*'[v] = s_*[v]$ *for all* $v \in \mathcal{V}$ *s.t.* $s_*[v]$ *defined, and* $s_*'[\overline{v}] = 0$.

Let us explain the semantics of the reformulation in Definition 4. By $X_\pi$ we denote the multiset of actions in a plan $\pi$. The variable $\overline{v}$ starts from the value 1 and switches to 0 when an action is applied that is not from the multiset $X = X_\pi$. Once a value 0 is reached indicating a deviation

## Algorithm 1 Iterative diverse planning scheme.

**Input:** Planning task $\Pi$, number of diverse plans $k$, number of total plans for search phase $K$, diversity metric $D$

    $P \leftarrow \emptyset$
    $\Pi' \leftarrow \Pi$
    **while** $|P| < K$ **do**
        $\pi \leftarrow$ some solution to $\Pi'$
        $P \leftarrow P \cup \{\pi' \mid \pi' \text{ is symmetric to } \pi\}$
        $X \leftarrow \bigcup_{\pi \in P} X_\pi$
        $\Pi' \leftarrow \Pi_X^-$ according to Definition 4
    **end while**
    **return** choose $k$ diverse plans from $P$, according to $D$

from plan $\pi$, it cannot be switched back to 1. Variables $\overline{v}_o$ encode the number of applications of the action $o$. The actions $o^r$ and $o^d$ are copies of the action $o$ in $X$ for the cases when $\pi$ is already discarded from consideration (variable $\overline{v}$ has switched its value to 0) and for discarding $\pi$ from consideration (switching $\overline{v}$ to 0), respectively. The latter happens if the action $o$ was already applied as many times as it appears in $X$. $o_i^f$ are copies of the action $o$ in $X$, counting the number of applications of $o$, as long as the number is not higher than the number of times it appears in $X$. These actions are applicable only while the plan is still followed. As mentioned above, ignoring plan reorderings sits well with the stability metric, but also with the uniqueness metric. For the state metric, note that although different reorderings of the same plan produce different sequences of states, these sequences will mostly be quite similar. Thus, we believe that it is more beneficial to spend the time on finding additional plans that are "set"-different instead of finding additional reorderings of the found plans. Note that in principle we could do both, if time permits.

When a set of plans is available, obtained, e.g., by applying structural symmetries (Shleyfman et al. 2015), one option would be to reformulate via a series of reformulations as in Definition 4. Another option is to forbid possibly more than just that set of plans by exploiting Definition 4 for forbidding a multiset of actions that is a superset of all plans in the set. In our implementation, we decided to follow the latter approach, depicted in Algorithm 1. Each iteration starts from the original task and forbids all plans found so far. In the last step, the algorithm selects a diverse subset of plans out of the set of plans found so far. In what follows, we discuss how such a selection can be done.

## 5.2 Selecting a Diverse Subset of Plans

The idea of selecting a set of plans in a post-processing phase is not new. A basic filtering and then clustering was performed over the set of plans for a top-k planning problem (Sohrabi et al. 2016; 2018). These approaches, however, may become time consuming when metric computation is computationally expensive. Hence, in this work, we instead apply a simple greedy algorithm, with a negligible computational overhead. We first order the found plans by their cost. Then, going from the cheapest plans to the more expensive ones, we find a pair of plans with the largest diversity score.

Starting with the found pair of plans, we iteratively construct the set by greedily choosing the next plan to add to the set, maximizing the diversity of the resulting set at that iteration step. We stop once the set reaches the requested size $k$. We note that the quality of the solution obtained by such an algorithm may be considerably improved. However, as we see next, even such a naive algorithm produces quite encouraging results.

## 6  Diversity-Bounded Diverse Planning

As previously mentioned, LPG-d as described by Nguyen et al. (2012) is a sound diversity-bounded diverse planner, although not complete. Similarly, our suggested approach can be used to produce a sound diversity-bounded diverse planner by post-processing the obtained plans differently. In general, such a post-processing procedure should find a collection of plans that adhere to certain constraints and that often corresponds to solving an NP-hard computational problem. For $D_{mma}$, that corresponds to finding a clique of size at least $k$, for a graph over vertices that correspond to plans found during the search phase and edges that correspond to pairs of plans of stability dissimilarity of at least $d$. Such cliques can be found using, e.g., mixed-integer linear program tools. In what follows, we use binary variables, one for each graph vertex to encode whether the vertex is a part of the selected clique. For each pair of vertices that are not connected by an edge, at most one of these vertices can belong to a clique. Thus, we introduce a constraint stating that if there is no edge between two vertices, then the sum of the two corresponding binary variables cannot exceed 1. An additional constraint requires the sum of all binary variables to be greater or equal to $k$, the number of the requested plans. Thus, valid assignments to the binary variables correspond exactly to cliques of size at least $k$. As a result, any optimization criteria can be chosen. Here, we choose to minimize the size of the obtained clique, finding a clique of size exactly $k$. This is done by minimizing the sum of all variables. Note that, while it is not required by the diversity-bounded diversity-bounded diverse planning problem, one can optimize other criteria while keeping the same set of constraints, and choosing a clique, e.g., maximizing the sum of pairwise stability measures.

## 7  Experimental Evaluation

In order to evaluate the feasibility of our suggested approach for deriving diverse sets of plans according to various existing metrics, we have implemented our approach on top of the Fast Downward planning system (Helmert 2006). Our planners, ForbidIterative (FI) diverse planners are available as part of the collection of ForbidIterative planners (Katz, Sohrabi, and Udrea 2019a). Further, we implemented an external component, that given a set of plans and a metric returns the score of the set under that metric (Katz and Sohrabi 2019).

We compare our approach for satisficing diverse planning to existing satisficing diverse planners, namely DLAMA planner (Bryce 2014), DIV (Coman and Muñoz-Avila 2011), $itA^*$, RWS, MQAd, MQAs, MQAtd, and MQAts

(Roberts, Howe, and Ray 2014), on state, stability, uniqueness, as well as a uniform linear combination over all subsets of these metrics, seven diversity metrics overall, shown in Table 1. Our diversity-bounded diverse planner is compared to the only existing diversity-bounded diverse planner LPG-d (Nguyen et al. 2012), on the $D_{mma}$ metric, varying the diversity parameter $d$ to obtain values $0.15$, $0.25$, and $0.5$ (see Table 2). We also varied the value of $k$, the number of required plans, for $k \in \{5, 10, 100, 1000\}$. For completeness, we include a comparison to LPG-d viewed as a satisficing diverse planner. To compare to all selected existing planners, we restrict our benchmark set to STRIPS domains with uniform action costs from the International Planning Competitions (IPC). This results in $1276$ tasks in $40$ domains.

The experiments were performed on Intel(R) Xeon(R) CPU E7-8837 @2.67GHz machines, with time and memory limits of 30min and 2GB, respectively. Our suggested approach iteratively solves a planning task, finds a set of plans, and creates a new task that forbids a superset of the plans found so far. Considering plans as multisets, ignoring the order between the actions, this superset is defined as the union of all plans found so far. Thus, we forbid reorderings of found plans, but also, possibly additional plans, corresponding to a union of multiple found plans. We are restricting the number of found plans to $1000$. For solving the (original and reformulated) planning tasks, we use an existing state-of-the-art agile planner. The planner that was chosen is *MERWIN* (Katz et al. 2018a). It performs a greedy best-first search (GBFS), alternating between four queues, novelty of the red-black heuristic, landmark count, preferred operators from the red-black heuristic, and preferred operators from the landmark count heuristic. The configuration has shown an exceptionally good performance on the IPC domains in our benchmark set (Katz et al. 2017). Note that while we report only results for MERWIN, we have also experimented with LAMA (Richter and Westphal 2010). The results were similar, therefore we report here only the results for MERWIN. A minor restriction in our choice of an external planner is the ability to work directly on SAS$^+$ representation, since our task reformulation is performed directly on SAS$^+$ and results in a SAS$^+$ task. This restriction is indeed somewhat minor, since most state-of-the-art planners do work on the grounded SAS$^+$ representation. In some cases, however, an adaptation might be required, since our implementation uses the input format of the Fast Downward planning system (Helmert 2006).

The solution to the computational problem of interest is chosen in the post-processing step from the found plans. Focusing first on satisficing diverse planning, if the desired number of plans $k$ is lower, we then greedily[1] choose a subset of size $k$ according to the given diversity metrics, as described in Section 5.2. Note that this can result in different subsets of plans chosen for different metrics. The algorithm is implemented as part of the external component (Katz and Sohrabi 2019). Each technique gets a score between 0 and 1 for each task and each metric, as described in previous

_______________
[1] We have experimented with exact techniques, based on mixed-integer linear programs, but found them to be prohibitively slow.

| | FI | DIV | DLAMA | itA* | LPG-0.15 | LPG-0.25 | LPG-0.5 | MQAd | MQAs | MQAtd | MQAts | RWS |
|---|---|---|---|---|---|---|---|---|---|---|---|---|
| **k=5** | | | | | | | | | | | | |
| coverage | **1143** | 95 | 178 | 611 | 705 | 701 | 680 | 277 | 499 | 2 | 615 | 51 |
| $Q_c$ | **1095.66** | 84.69 | 127.26 | 539.13 | 527.10 | 526.46 | 488.91 | 190.37 | 447.37 | 1.80 | 533.03 | 39.25 |
| $D_a$ | **736.88** | 33.65 | 123.07 | 271.77 | 402.72 | 412.39 | 455.86 | 213.83 | 277.70 | 0.40 | 322.17 | 30.62 |
| $D_s$ | **585.34** | 45.01 | 96.35 | 200.58 | 321.62 | 322.02 | 336.41 | 144.86 | 143.73 | 1.09 | 229.62 | 25.32 |
| $D_u$ | **1093.70** | 53.10 | 176.00 | 527.70 | 688.10 | 689.10 | 671.90 | 275.20 | 486.70 | 0.60 | 539.40 | 41.20 |
| $D_s\,D_a$ | **640.46** | 39.33 | 108.60 | 236.17 | 362.13 | 367.17 | 396.03 | 179.34 | 210.71 | 0.74 | 275.90 | 27.97 |
| $D_s\,D_u$ | **837.18** | 49.06 | 136.19 | 364.14 | 504.81 | 505.55 | 504.14 | 210.03 | 315.21 | 0.85 | 384.51 | 33.26 |
| $D_u\,D_a$ | **915.87** | 43.37 | 149.50 | 399.74 | 545.39 | 550.74 | 563.87 | 244.51 | 382.20 | 0.50 | 430.79 | 35.91 |
| $D_a\,D_u\,D_s$ | **791.81** | 43.92 | 131.11 | 333.35 | 470.75 | 474.45 | 487.97 | 211.30 | 302.71 | 0.70 | 363.73 | 32.38 |
| **k=10** | | | | | | | | | | | | |
| coverage | **1113** | 1 | 133 | 422 | 661 | 652 | 610 | 168 | 398 | 0 | 430 | 31 |
| $Q_c$ | **1060.08** | 0.93 | 92.43 | 376.64 | 508.15 | 500.38 | 433.88 | 106.96 | 361.55 | 0.00 | 363.00 | 23.68 |
| $D_a$ | **681.08** | 0.48 | 88.79 | 191.66 | 384.80 | 394.07 | 418.92 | 136.97 | 222.69 | 0.00 | 219.30 | 21.38 |
| $D_s$ | **534.93** | 0.52 | 71.32 | 164.53 | 300.97 | 302.22 | 307.07 | 93.88 | 114.76 | 0.00 | 175.29 | 15.84 |
| $D_u$ | **1054.53** | 1.00 | 132.62 | 353.76 | 648.07 | 645.27 | 608.53 | 167.60 | 394.91 | 0.00 | 365.40 | 28.91 |
| $D_s\,D_a$ | **590.98** | 0.50 | 79.38 | 178.10 | 342.86 | 348.10 | 362.80 | 115.43 | 168.73 | 0.00 | 197.29 | 18.61 |
| $D_s\,D_u$ | **792.08** | 0.76 | 101.92 | 259.14 | 474.51 | 473.71 | 457.80 | 130.74 | 254.84 | 0.00 | 270.34 | 22.37 |
| $D_u\,D_a$ | **868.10** | 0.74 | 110.68 | 272.71 | 516.42 | 519.66 | 513.73 | 152.28 | 308.80 | 0.00 | 292.35 | 25.15 |
| $D_a\,D_u\,D_s$ | **745.40** | 0.67 | 97.09 | 236.65 | 444.59 | 447.13 | 444.71 | 132.82 | 244.12 | 0.00 | 253.33 | 22.04 |
| **k=100** | | | | | | | | | | | | |
| coverage | **909** | 0 | 11 | 37 | 550 | 535 | 433 | 32 | 170 | 0 | 78 | 15 |
| $Q_c$ | **849.21** | 0.00 | 7.28 | 31.80 | 450.36 | 431.24 | 296.07 | 17.41 | 165.39 | 0.00 | 66.48 | 11.61 |
| $D_a$ | **492.55** | 0.00 | 6.89 | 22.12 | 339.17 | 337.18 | 319.36 | 27.02 | 100.35 | 0.00 | 51.24 | 11.53 |
| $D_s$ | **404.63** | 0.00 | 5.78 | 17.90 | 262.98 | 258.64 | 227.79 | 18.96 | 57.49 | 0.00 | 34.09 | 7.70 |
| $D_u$ | **834.18** | 0.00 | 11.00 | 32.70 | 548.63 | 534.11 | 432.95 | 31.99 | 169.99 | 0.00 | 76.74 | 14.95 |
| $D_s\,D_a$ | **438.70** | 0.00 | 6.29 | 20.01 | 301.04 | 297.85 | 273.43 | 22.99 | 78.92 | 0.00 | 42.67 | 9.61 |
| $D_s\,D_u$ | **617.02** | 0.00 | 8.39 | 25.30 | 405.77 | 396.35 | 330.36 | 25.48 | 113.74 | 0.00 | 55.42 | 11.33 |
| $D_u\,D_a$ | **661.18** | 0.00 | 8.94 | 27.41 | 443.91 | 435.63 | 376.15 | 29.51 | 135.17 | 0.00 | 63.99 | 13.24 |
| $D_a\,D_u\,D_s$ | **569.55** | 0.00 | 7.86 | 24.24 | 383.55 | 376.58 | 326.60 | 25.99 | 109.28 | 0.00 | 54.02 | 11.39 |
| **k=1000** | | | | | | | | | | | | |
| coverage | **552** | 0 | 0 | 0 | 406 | 363 | 234 | 0 | 0 | 0 | 0 | 7 |
| $Q_c$ | **543.14** | 0.00 | 0.00 | 0.00 | 361.52 | 313.51 | 170.68 | 0.00 | 0.00 | 0.00 | 0.00 | 5.96 |
| $D_a$ | **263.58** | 0.00 | 0.00 | 0.00 | 244.58 | 224.01 | 174.19 | 0.00 | 0.00 | 0.00 | 0.00 | 5.38 |
| $D_s$ | **206.54** | 0.00 | 0.00 | 0.00 | 194.19 | 173.49 | 118.10 | 0.00 | 0.00 | 0.00 | 0.00 | 3.66 |
| $D_u$ | **490.44** | 0.00 | 0.00 | 0.00 | 405.93 | 362.92 | 234.00 | 0.00 | 0.00 | 0.00 | 0.00 | 7.00 |
| $D_s\,D_a$ | **233.82** | 0.00 | 0.00 | 0.00 | 219.35 | 198.71 | 146.10 | 0.00 | 0.00 | 0.00 | 0.00 | 4.52 |
| $D_s\,D_u$ | **348.47** | 0.00 | 0.00 | 0.00 | 300.05 | 268.20 | 176.05 | 0.00 | 0.00 | 0.00 | 0.00 | 5.33 |
| $D_u\,D_a$ | **375.79** | 0.00 | 0.00 | 0.00 | 325.25 | 293.45 | 204.11 | 0.00 | 0.00 | 0.00 | 0.00 | 6.20 |
| $D_a\,D_u\,D_s$ | **318.92** | 0.00 | 0.00 | 0.00 | 281.54 | 253.44 | 175.40 | 0.00 | 0.00 | 0.00 | 0.00 | 5.35 |

Table 1: Overall summed scores for various metrics, for k=5, 10, 100, and 1000. $D_a$ stands for *stability*, $D_s$ for *state*, and $D_u$ for *uniqueness* diversity metrics. Rows that correspond to a linear combination of diversity metrics are marked with all combined metrics. $Q_c$ stands for *cost* quality metric. Best results are highlighted in bold.

sections. If not enough unique plans were found by some planner on a task, the planner gets the score of 0 for that task. Table 1 depicts the summed scores for all planners on all metrics, for various values of $k$, from 5 to 1000. First, note that our approach excels on all metrics, for both diversity and quality. This is due in part to an increased coverage, by 62% for $k = 5$, 68% for $k = 10$, 65% for $k = 100$, and 36% for $k = 1000$. However, as we later see, that is not the only source of improved performance. For the quality metric, we improve by over 100% for the smaller values of $k = 5$ and 10, by 88% for $k = 100$, and by 50% for $k = 1000$. For various diversity metrics, the improvement is between 59% and 74% for $k = 5$ and 10, and between 45% and 54% for $k = 100$. For $k = 1000$, the improvement is much more modest: 6% for the *state* metric, 8% for the *stability* metric, and 21% for the *uniqueness* metric. Note that while for smaller $k$ values there are several techniques that are somewhat comparable in their performance to ours, larger $k$ values seem to be challenging for most techniques.

The only exception is *LPG-d*, which performs rather well even for large $k$ values. In fact, despite solving a different computational problem, *LPG-d* is the strongest competitor to our approach for all tested values of $k$.

In order to go beyond the aggregated results, Figure 2 shows the comparison between our technique and *LPG-d* with $d = 0.5$, the best performing contestant for $k = 5$. The plots show two diversity metrics, *stability* and *state*. Each task corresponds to a single point, with coordinates representing the metric value. All points above the diagonal are in favor of *LPG-d*, and below the diagonal are in favor of our technique. The points on the axes correspond to tasks that either were solved by one technique but not the other or the score obtained by one of the techniques was 0. For the metric stability in Figure 2(a), there are 884 tasks below the diagonal, 490 of these tasks are on the x axis. There are 286 tasks above the diagonal, 41 of these tasks are on the y axis. For the metric state in Figure 2(b), there are 938 tasks below the diagonal, with 495 tasks on x axis and 237

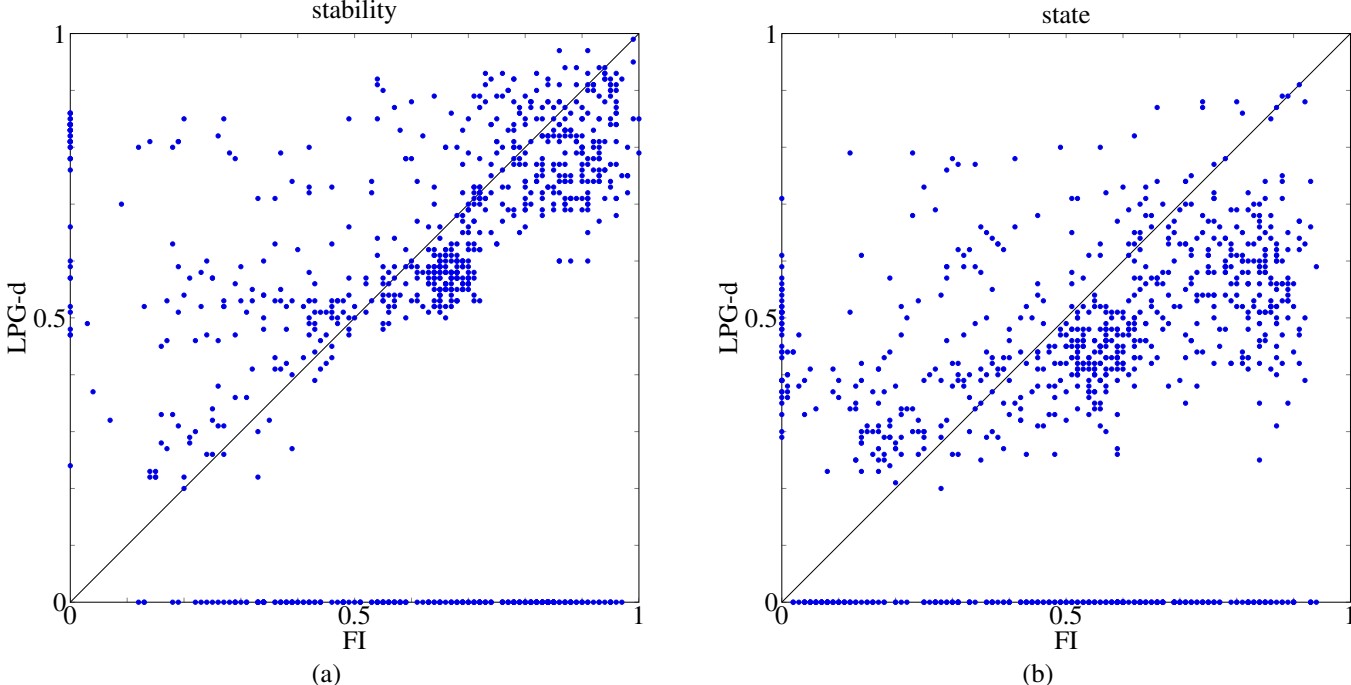

Figure 2: Comparison of our technique (FI) to the LPG-d planner with $d = 0.5$ on (a) $D_a$ and (b) $D_s$ metrics for $k = 5$.

| k | 0.15 | | | 0.25 | | | 0.5 | | |
|---|---|---|---|---|---|---|---|---|---|
| | **bFI** | **LPG-d** | Dom | **bFI** | **LPG-d** | Dom | **bFI** | **LPG-d** | Dom |
| 5 | **1011** | 675 | 28/6 | **974** | 671 | 25/8 | **890** | 652 | 24/10 |
| 10 | **946** | 632 | 26/7 | **912** | 623 | 27/9 | **771** | 586 | 25/10 |
| 100 | **569** | 532 | 17/13 | 454 | **517** | 15/15 | 213 | **433** | 8/16 |
| 1000 | 152 | **396** | 7/17 | 80 | **359** | 3/18 | 3 | **234** | 1/15 |

Table 2: Comparison of bounded-diversity score (total number of solved tasks) for k=5, 10, 100, and 1000 for the *stability* metric ($D_{mma}$). Best results are bolded. Dom shows # of domains with superior performance for bFI/LPG-d.

tasks above the diagonal, with 39 tasks on y axis. Observe that most of the tasks are not near the diagonal, and thus these techniques are rather complementary. We note that for *FI*, the score was computed with a greedy algorithm. Exact solutions, although slower, might have got a better score.

Moving now to diversity-bounded diverse planning, we increased the bound on the number of plans found in the first phase to 2000, to give the planner some choice for $k = 1000$. The solution here is obtained by solving the binary linear program, as described in Section 4.2 with the CPLEX solver in its default configuration. The implementation is available as part of the external component (Katz and Sohrabi 2019). While in general these programs have up to 2K binary variables and up to 4M constraints, we observe that the run time of the solver is rarely above 10 seconds, with the peak reaching 47 seconds. If binary linear program was solved by the solver (feasible solution found), the planner gets 1, and otherwise (infeasible) 0. We post-process the set of plans from both our approach and LPG-d in the same way. Ta-

ble 2 shows the overall summed scores over all instances, as well as the number of domains where each approach exhibits superior performance. As a reminder, our approach chooses $k$ plans out of the found plans with $D_{mma}$ above the given threshold. Thus, our approach has a clear disadvantage when there is little or no choice, as in the case of the largest $k$ values in our experiment. For smaller $k$ values ($k = 5, 10$), there is a clear advantage to our approach, for all tested bounds on $D_{mma}$.

## 8 Summary and Future Work

We have presented various diverse planning computational problems and classified the existing diverse planners with their respective problems. Key contributions of this paper include: (1) characterization of optimal, bounded, and satisficing diverse planning problem; (2) introducing an external validation component for diverse planning; (3) addressing the satisficing and bounded-diversity diverse planning problems by iteratively solving a modified planning task using existing classical planners, escaping the need to adapt a planner to each new diversity metric. We have empirically demonstrated the benefits of using such an approach, considerably improving the state-of-the-art in satisficing diverse planning and favorably competing with the state-of-the-art in bounded-diversity diverse planning.

For future work, in satisficing diverse planning, we intend to explore alternative ways of reformulating a planning task, aiming at tackling a specific diversity metric. For various optimal diverse planning computational problems, it is often not clear how to create a non-trivial planner for that problem at all. For example, an optD-k optimization problem,

requires to generate a set of plans that is diversity-optimal. A naive solution might require generating all possible plans first, which might be infeasible, especially in cases when the set of all plans is infinite. Focusing on such planning problems is a promising research direction.

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
