# OpenReview forum: "Reshaping Diverse Planning: Let There Be Light!"
_icaps-conference.org/ICAPS/2019/Workshop/HSDIP_

### Official Review · AnonReviewer1 · 2019-03-31
**well-written paper with interesting theoretical and empirical contents**

**Rating:** 9
**Confidence:** 4

**Review:**

The paper formally introduces several variants of diverse planning and
proposes a new algorithm for solving the satisficing and
bounded-diversity variants.

I enjoyed reading the paper. It clearly explains existing work on the
topic of diverse planning, characterizes the existing approaches with
respect to new formal definitions of diverse planning variants,
introduces a simple way of finding diverse plans and shows that the new
approach works well for the IPC benchmark tasks.

I think having an independent program for calculating plan diversity is
very useful. I was therefore surprised to see that the paper only
promises to make it available "upon request". Would you consider making
the tool publically available, preferably at a location that guarantees
persistency and citability like Zenodo? What about the source code of
the planner?

Other than that, I just have minor comments and typos for the authors.


Introduction:

Few examples -> Examples

plans quality -> plan quality

plans from among these -> plans among these

as well as ++the++ diverse planner construction paradigm


Definition 1:

all \Pi's plans -> all plans for \Pi


Bounded-diversity Diverse Planning

a NP-hard -> an NP-hard

represent++s++ a clique


Experimental Evaluation

an uniform -> a uniform

We are restricting the --size-- ++number++ of found plans

result++s++ for MERWIN

binary linear --problem-- ++program++


References:

Some page numbers are missing.

---

> ### Author Response · Authors · 2019-04-10
> **We thank the reviewers for their constructive feedback.**
>
> We are making efforts toward having the code publicly available as soon as possible.

---

### Official Review · AnonReviewer2 · 2019-04-04
**Interesting approach to an interesting problem**

**Rating:** 7
**Confidence:** 4

**Review:**

The paper addresses the issue of Diverse Planning, that is, the need for finding sets of plans which are sufficiently different among themselves. This is relevant and several distinct applications have been explored in previous literature. The paper makes a number of contributions, most relevantly:
(1) a taxonomy of the different computational problems related to plan diversity that are targeted in previous literature;
(2) the adaptation of a previous “task reformulation” approach from a slightly different context to the computation of sets of diverse plans; and
(3) a throrough empirical analysis of the approach (2).

Concerning (1), several problems are formally defined that subsume the somewhat different problems tackled in previous work. To increase readability, I would suggest the authors that they include in the definitions they present here the relevant numeric parameters. Let me clarify with an example, the definition of the “sat-k” problem:

sat-k : Given k, find a diverse planning solution.

It reads a bit confusing that “k” is not present in the definition of the problem. Of course one can understand what the authors mean by the context, but I think it could be much clearer to instead talk of “k-diverse planning solutions”, and then simply state that the sat-k problem is “Given k, find a k-diverse planning solution”, or something along these lines. This applies to most of the definitions in Section 4.

Regarding (2), the authors propose an iterative approach that takes a planning task \Pi, computes a plan \pi, and then creates a second task based on both \Pi and \pi that disallows solutions similar to \pi. This is interesting because (in contrast with most previous techniques) it allows the use of any planner for the diverse planning task. However, the approach is essentially the same as the “plan forbid” reformulation in (Katz et al. 2018b), only that for a slightly different objective, and I think this should be stated more clearly. It would also be necessary to discuss the properties (e.g. completeness) of the approach. As far as I understand, roughly the reformulation forbids not only the already-found plans, but also any other plan that includes only operators from previous plans, but where the count of these operators is smaller than the aggregated count in all previous plans. I am not sure that this results in a complete diverse-planner, as the approach might be pruning too many feasible plans. Also, it seems that the plans that are pruned could indeed be better than the ones already found? Consider for instance a task where the only 2 plans are <o1> and <o1, o1>. If the loop in Algorithm 1 finds the second plan first, then it looks like the first one will never be found. If the problem demands that a set of k=2 diverse plans be found, I am not sure how the presented approach will cope with that. Additionally, finding the bad plan first disallows finding the good one ever, which is not that nice in terms of quality.

Also concerning Algorithm 1, perhaps the authors could clarify or give an intuition in the text about what is the role of symmetries within the algorithm.

Finally, concerning (3), the paper includes a thorough empirical comparison of the proposed approach, dubbed “FI”, to other previous diverse-planning techniques. FI systematically outperforms the previous techniques by a large margin, scaling up much better than most of them.

The subject of the paper is relevant and well within the scope of the workshop; the proposed technique is an adaptation of the technique presented by Katz et al. (2018b) to a slightly different context, and the empirical study is interesting enough and deserves to be presented and discussed. I have presented a few concerns above, but I think that in the context of a workshop, there’s enough time to clarify them before the presentation. I thus recommend acceptance.


A number of minor comments and suggested edits follow:
* [abstract] “classifying the existing planners to these problems” --> classifying the existing planners according to this taxonomy? (just a suggestion)
* [p.1] “Few examples include malware detection...” ---> “Some examples”
* [p.7] "Our diversity-bounded diverse planner is compared to the only existing diversity-bounded diverse planner LPG-d" --> it is not clear in the text that this refers to Table 2?

---

> ### Author Response · Authors · 2019-04-10
> **We thank the reviewers for their constructive feedback.**
>
> Indeed, the approach, as presented in Section 5, is not complete.
> If completeness is desired, it can be obtained by, e.g., generating for each found plan $\pi$ all plans $\pi'$ with $X_{\pi'} \subseteq X_{\pi}$. A minor adaptation of an algorithm for generating all reorderings of a given plan, presented in Katz et al. 2018b, would allow for generating such plans.
>
> The symmetries are not imperative to the approach and are used for generating additional plans from previously found plans, and thus possibly reducing the number of required iterations.

---

### Meta-Review · Program_Chairs · 2019-04-25

**Recommendation:** Accept
**Confidence:** 5

**Metareview:**

Dear Authors,
thank you very much for your submission. We are happy to inform you that
we have decided to accept it and we look forward to your talk in the workshop.
Please, go over the feedback in the reviews and correct or update your papers
in time for the camera ready date (May 24).
Best regards
HSDIP organizers